# Effect of Feed Containing *Hermetia illucens* Larvae Immunized by *Lactobacillus plantarum* Injection on the Growth and Immunity of Rainbow Trout (*Oncorhynchus mykiss*)

**DOI:** 10.3390/insects12090801

**Published:** 2021-09-07

**Authors:** Dooseon Hwang, Chae-Hwan Lim, Seung Hun Lee, Tae-Won Goo, Eun-Young Yun

**Affiliations:** 1Department of Integrative Biological Sciences and Industry, Sejong University, Seoul 05006, Korea; h.michael8837@gmail.com (D.H.); hwanni995@naver.com (C.-H.L.); g-d-lsh@hanmail.net (S.H.L.); 2Department of Biochemistry, School of Medicine, Dongguk University, Gyeongju 38066, Korea; gootw@dongguk.ac.kr

**Keywords:** *Hermetia illucens*, *Oncorhynchus mykiss*, insect, antimicrobial peptide, artificial feed, immunity, growth

## Abstract

**Simple Summary:**

In this study, we evaluated the effect on the growth and immunity of rainbow trout of a feed formulated using *Hermetia illucens* with increased antimicrobial peptides expression by *Lactobacillus plantarum* infection (ImHIL). As a result, growth and immunological indicators improved, and therefore, ImHIL is expected to become a good feed source for rainbow trout aquaculture.

**Abstract:**

We investigated the effects of a feed containing *Hermetia illucens* larvae injected with bacteria on the growth and immunity of *Oncorhynchus mykiss*. The feed was prepared by replacing fishmeal in feed with 25 and 50% nonimmunized (HIL25, HIL50) or immunized HIL (ImHIL25, ImHIL50), and its protein:fat:carbohydrate ratio was 45:15:18. ImHIL extracts showed inhibitory activity against fish pathogenic bacteria. Both red blood cell count and insulin-like growth factor-1 as the growth indicator were the highest among the groups at week 6 after feeding in the ImHIL50 group. As immune indicators, blood aspartate aminotransferase levels were lower in the ImHIL25 and ImHIL50 groups than in that of other groups at week 6 after feeding, and lysozyme content was significantly higher in ImHIL25 and ImHIL50. The above results demonstrate that ImHIL has a beneficial effect on the improvement of growth and immunity. Accordingly, we suggest that ImHIL has the potential to be a good feed source in aquaculture.

## 1. Introduction

Fish are considered an excellent nutrition source because of their low-calorie content, balanced protein value, and high unsaturated fatty acid content [1]. In addition, fish consumption increased sharply worldwide due to population growth and the increasing demand for high-quality protein foods. Thus, aquaculture is essential for a sustainable fish supply. Fishmeal and fish oil were mainly used as raw materials in feed manufacturing, one of the most important aquaculture aspects [2]. However, these ingredients were recently decreasing in supply; therefore, discovering new raw materials to replace them is of importance [3,4]. Soybean, rapeseed, sunflower, spirulina, yeast, food waste, and animal byproducts were considered as alternative ingredients for fishmeal replacement [5]. However, these raw materials have problems, including low fish growth rates, uneven nutrient content, environmental pollution, and competition for use with humans [6]. Furthermore, carnivorous fish require 10–20% more protein in their diet than phytophagous fish [7]. Therefore, it is essential to find protein-rich raw materials that can replace fishmeal to develop alternative feeds in aquaculture.

Since insects are considered a good protein source [8], several studies showed that they have the potential to replace fishmeal in aquaculture feeds. Insects such as mealworms, silkworms, house fly larvae [9], black soldier fly (*Hermetia illucens*) larvae (HIL) [2,10,11], and crickets [12] were studied as fishmeal substitutes. In particular, HIL has a relatively high protein content (41%) [8] and was reported to be a promising resource for fishmeal replacement because of its amino acid composition profile, which is similar to that of fishmeal [13]. HIL has advantages in terms of organic waste consumption, easy mass production, space-saving, and eco-friendliness [14]. HIL was reported to have a positive effect on growth when it replaces fishmeal in fish diets for species such as turbot (*Psetta maxima*), Nile tilapia (*Oreochromis niloticus*), mirror carp (*Cyprinus carpio*), yellow catfish (*Pelteobagrus fulvidraco*), rainbow trout (*Oncorhynchus mykiss*), Eurasian perch (*Perch fluviatilis*), and European seabass (*Dicentrachus labrax*) [10,11,15,16,17,18,19,20,21]. HIL-containing fish feed downregulates the expression of inflammatory factors such as toll-like receptor 22 (TLR22), CCAAT-enhancer-binding protein (C/EBPβ), mitogen-activated phosphokinase p38 (p38 MAPK), lipoxygenase-5 (LOX5), and oxidative stress-related genes, such as glutathione peroxidase enzyme (Gpx1), copper/zinc-superoxide dismutase (Cu/Zn-SOD), and manganese superoxide dismutase (Mn-SOD) [2].

Insects produce the most diverse antimicrobial peptides (AMPs) for defense [22], of which stick insects (*Peruphasma schultei*) and HIL were found to have more than 50 different AMP genes [23,24,25]. Antimicrobial peptides are among the defense mechanisms of eukaryotes with direct activity against a wide range of pathogens [17], including bacteria, yeast, and fungi [18]. In addition to their antimicrobial effect, AMPs positively affect growth and immune responses, including increased immunity in pigs [22], improved chicken growth and meat quality, and increased fish growth rate [7,17,26]. When fed diets containing AMPs, the height and surface area of the intestinal villi increase, thereby increasing nutrient absorption in the intestines, and in turn, improving growth performance [23]. Although synthetic antibiotics increase fish growth rate, residual antibiotics cause side effects such as antibiotic resistance [27,28]. However, AMP, a natural antibacterial agent derived from organisms, is safe in the environment and body [22]. Recently, a study on the effect of rainbow trout diets containing HIL on growth performance, feed efficiency, nutrient deposition, blood glucose, lipid digestibility, and sensory profile was conducted [17,18,19,20,21]. Similarly, studies were conducted on effects on broiler growth performance, feed choice, blood traits, carcass characteristics, and meat quality [14,29]. However, to date, there were no studies on the effect of feed containing substantial amounts of AMPs on enhancing fish immunity.

Therefore, in this study, the effect of feed prepared using immunized HIL (ImHIL), which contains a large amount of various natural AMPs in the hemolymph owing to injection with bacteria, on the growth and immunity of rainbow trout was investigated. Since the maximum expression of AMP in the insect body can be induced by septic injury with bacteria [30], we evaluated the efficacy of diets containing HIL immunized by septic injury on rainbow trout (*O. mykiss*). We compared the effects on growth performance and immunity of *O. mykiss* after preparing and feeding diets in which 25 and 50% of the existing fishmeal was replaced with immunized or nonimmunized HIL.

## 2. Materials and Methods

### 2.1. HIL, Bacteria Strain, and Immunization

The last-instar HIL and ImHIL used in feed formulation were purchased from Agrogreen (Gongju, Korea). *Lactobacillus plantarum* (KACC 10552), used to induce AMPs in HILs, was obtained from the Korean Agricultural Culture Collection (KACC).

After fasting for one day, a sterilized 1-mm diameter pin was soaked in 1 × 10^10^ CFU *L. plantarum*, and then injected into the side of the HIL’s abdomen to induce the expression of AMPs. Injected HILs were left for 6, 12, 24, 36, 48, and 72 h at room temperature without feeding. Subsequently, the HILs were rapidly frozen in liquid nitrogen and ground. Afterward, the HIL powder was used to produce rainbow trout feed. The antibacterial activity of the ImHIL extract was analyzed using *Edwardsiella tarda* and *Vibrio harveyi*, which are gram-negative bacteria pathogenic to fish.

### 2.2. Preparation of HIL and ImHIL Extract

The HIL and ImHIL powders were mixed with 20% acetic acid (AcOH) at a ratio of 1:10 (*w*/*v*), boiled for 30 min, and centrifuged at 8935× *g* for 30 min. The supernatant was filtered through a 0.45 μm syringe filter. Subsequently, the supernatant was concentrated using a speed vacuum (EYELA, Tokyo, Japan) for 48 h, and the quantity of extract was measured. The concentrated extract was resuspended in 0.05% AcOH and used for antimicrobial activity assays.

### 2.3. qRT-PCR for AMP Expression Analysis in ImHIL

To confirm the maximum expression timing of AMPs in the body of ImHIL, we performed quantitative real-time PCR (qRT-PCR) and a radial diffusion assay (RDA). Total RNA from ImHIL or HIL was separated using TRIzol^®^ Reagent (Invitrogen, Carlsbad, CA, USA). cDNA was synthesized using a Quantinova^®^ reverse transcription kit (Qiagen, Hilden, Germany). HIL and ImHIL AMPs (cecropin 1 and defensin) were determined using the SensiFAST™ Sybr No-Rox Mix, 2× (Bioneer, Daejeon, Korea). Actin was used as an endogenous control, and relative gene expression levels were analyzed using relative quantitation. The primers used are listed in Table 1.

### 2.4. RDA and MIC for the Analysis of the Antibacterial Activity of ImHIL

The ImHIL extracts were used for RDA against *E. tarda* and *V. harveyi*. Underlay agar (10 mL; 100 mM Na-ⓟ/citrate buffer, TSB, agarose, distilled water) was mixed with pathogenic bacteria. When the underlay agar hardened, a 3-mm diameter hole was punctured into the agar, and 5 µL of the ImHIL extract was loaded into the hole. After 3 h of incubation, 10 mL of overlay agar was used to cover the hole and incubated for 18 h. To measure the minimal inhibitory concentration (MIC), 1 × 10^5^ cells of *E. tarda and V. harveyi* were seeded in each well of a 96-well plate with ImHIL extracts. After 18 h, the OD_600_ value was measured.

### 2.5. Formulated Feed Manufacture and Analysis of the Proximate Composition of the Feeds

The raw materials of the rainbow trout feed used were HIL/ImHIL/fishmeal powder, α-starch, mineral mixture, vitamin mixture, and wheat flour. We formulated the following five artificial feeds according to AMPs-induction and HIL/ImHIL content: HIL0 (HIL 0%), HIL25 (25% HIL based on fishmeal, nonimmunized), ImHIL25 (25% HIL based on fishmeal, immunized), HIL50 (50% HIL based on fishmeal, nonimmunized), and ImHIL50 (50% HIL based on fishmeal, immunized). Based on each raw materials composition, the rainbow trout feed was formulated by setting each raw material’s ratio to 45% protein, 15% fat, 18% carbohydrate, and 14% ash [19,31]. All ingredients and 60% distilled water were kneaded using a mixer, pulled using a 6-mm diameter mincer, and dried at room temperature for 48 h. The dried feed was cut into 3-mm lengths to prepare pellets and used in the experiment. The pellets were stored at −20 °C and then at 4 °C the day before use. Proximate analysis of the five artificial diets was carried out according to the standard methods [32]. Total nitrogen and crude protein were analyzed using a FossTecator digestion system (FOSS, Hilleroed, Denmark) and a Vadopest 50s automatic nitrogen quantitative analyzer (Gerhardt Analytical Systems, Königswinter, Germany). Ether extract was determined using an ST 243 Soxtec solvent extraction system (Fisher Scientific, Hempton, NH, USA). The analysis of the ash was carried out as follows. After weighing the crucible, the sample was weighed, transferred to an incinerator, and heated at 550–600 °C for 2 to 3 h; this continued until white to grayish-white ash was obtained. After incineration, it was transferred to a desiccator, cooled, and immediately weighed when it reached room temperature to calculate the amount of ash of the sample. The carbohydrate content was calculated by excluding the content of crude protein, crude fat, moisture, and ash from the rainbow trout feed weight.

### 2.6. Fish, Rearing Conditions, and Sampling

The *O. mykiss* used in this experiment were purchased from Donghae STF (Yeongwol, Korea). The purchased rainbow trout was triploid, female, and was in the juvenile stage. The initial body weight and length of the fish were measured individually. Five artificial diet feeding groups of HIL0, HIL25, HIL50, ImHIL25, and ImHIL50 were treated in triplicate, and a total of 15 tanks were used. Eleven *O. mykiss* were then randomly introduced into one of 15 tanks (300 L) filled with filtered freshwater (temperature: 19.12 ± 0.23 °C, pH: 8.03 ± 0.06, dissolved oxygen: 7.95 ± 0.25 mg L^−1^). The photoperiod maintained 12L/12D and aerated continuously, and the fish breeding density was 6.43 ± 0.25 kg m^−3^. The five artificial diets were fed daily for 6 weeks [33,34]; at weeks 3 and 6, all fish in the tank were weighed and their lengths were measured. Afterward, three fish per tank were sacrificed to collect liver and blood samples. Mortality was recorded daily.

### 2.7. Analysis of Somatic and Hematological Indices

At weeks 3 and 6 after the feeding experiment, following a 1-d fasting period, all fish weights and lengths were measured individually, and the average weights and lengths were obtained. Three fish per tank (nine fish per group) were sacrificed using tricaine methanesulfonate (MS-222; Sigma-Aldrich, St Louis, MO, USA). A blood sample was taken from the tail vein using a syringe and heparinized vacuum tube. Hematologic parameters (packed cell volume and red blood cell count) were measured using whole blood. Serum was isolated from whole blood. Whole blood was allowed to stand overnight at 4 °C, followed by centrifugation at 1000× *g* for 15 min. The supernatant was used to measure insulin-like growth factor-1 (IGF-1), aspartate aminotransferase (AST), lysozyme activity, and immunoglobulin M (IgM). After blood collection, the same fish’s liver was surgically removed and stored at −20 °C for hepatosomatic index (HSI) determination.

#### 2.7.1. Red Blood Cell (RBC) Count

RBC count was determined using an optical microscope (BX51; Olympus, Tokyo, Japan), and serum was measured after a 200-times dilution. The RBC count was measured in five parts, including four parts on the edge of the hemocytometer (HSU-1400; Paul Marienfeld, Lauda–Königshofen, Germany) and one in the middle, and the final number of RBCs was determined.

#### 2.7.2. Packed Cell Volume (PCV)

PCV was measured immediately after sampling using a microhematocrit centrifuge (DSC-100MH-3; Digisystem Laboratory Instruments, New Taipei City, Taiwan). After stirring the blood in a heparin tube for 5 s, the 7.5 cm capillary was filled 5 cm blood sample, and one end was closed with wax plate. The blocked side was placed outside the rotor and centrifuged at 12,000× *g* for 15 min. Subsequently, the entire length and the RBC length were measured.

#### 2.7.3. Insulin-Like Growth Factor-1 (IGF-1)

IGF-1 levels in *O. mykiss* serum were measured using an enzyme-linked immunosorbent assay (ELISA) kit (Fish Insulin-like growth factors 1, IGF-1 ELISA kit, Cusabio, Houston, TX, USA) and analyzed according to the manufacturer’s instructions.

### 2.8. Analysis of Immunity Enhancement

Blood serum was used to assess the immunity-enhancing parameters.

#### 2.8.1. Aspartate Aminotransferase (AST)

AST activity in the serum was measured using a commercial analysis kit (Asan Pharm. Co., Ltd., Seoul, Korea) according to the manufacturer’s instructions.

#### 2.8.2. Lysozyme Activity

*O. mykiss* blood serum lysozyme activity was measured using an Enzchek™ Lysozyme Assay kit (Invitrogen, Carlsbad, CA, USA) according to the manufacturer’s instructions.

#### 2.8.3. Immunoglobulin M (IgM)

IgM in the blood serum of *O. mykiss* was measured using an enzyme-linked immunosorbent assay (ELISA) kit (Fish immunoglobulin M (IgM) ELISA kit, Cusabio, Houston, TX, USA) and analyzed according to the manufacturer’s instructions.

## 3. Results and Discussion

### 3.1. Establishment of Conditions for Inducing the Maximum Amount of AMPs in ImHIL

AMPs expression in ImHILs injected with *L. plantarum* was determined using qRT-PCR. The expression level of cecropin 1 was 27-fold higher than that in the control group at 24 h after bacterial injection, decreasing thereafter. The maximum expression time of defensin was also 24 h after injection, and at this time, the expression level was 23-fold higher than in the control (Figure 1A). These results were consistent with previous studies where the maximum expression of AMP in *L. casei*-injected HIL, the same genus as *L. plantarum*, was 24 h postinjection [35,36]. Therefore, in this study, ImHIL at 24 h after the injection of bacteria, at the maximum expression of cecropin 1 and defensin, was used as feed ingredients for rainbow trout.

After preparing the ImHIL extract under the aforementioned conditions, the antibacterial activity against two types of pathogenic bacteria causing disease in fish was tested. Accordingly, the inhibition zone’s size against *E. tarda*, a pathogenic bacterium that causes sepsis in fish, treated with 0, 500, and 1000 μg/5 μL of ImHIL extract grew to 0.3, 1.1, and 1.9 cm, respectively, in a concentration-dependent manner. On the other hand, for *V. harveyi*, a pathogenic bacterium causing hepatic bleeding, the size of the bacterial growth inhibitory region increased to 0.3, 1.3, and 1.7 cm, respectively, using the same concentrations as for *E. tarda* in a concentration-dependent manner. Therefore, the ImHIL extract was observed to have antimicrobial activity against both *E. tarda* and *V. harveyi* (Figure 1B). These results are thought to be because various AMPs are induced when bacteria invade the insect hemolymph, and they generally exhibit a broad spectrum of antibacterial activity [22]. In addition, the MIC of the HIL extract against *E. tarda* and *V. harveyi* was 1 × 10^5^ CFU/200 μg (Figure 1C).

### 3.2. Ingredient of ImHIL-Containing Feed

To investigate the effect of feed containing ImHIL on the growth and immunity of rainbow trout, we formulated five artificial feeds according to the induction of AMPs and HIL content: HIL0, HIL25, HIL50, ImHIL25, and ImHIL50. All five feeds were manufactured after calculating the protein:carbohydrate:fat ratio to be 45:15:18 based on the raw materials’ nutrient composition.

Based on the proximate analysis, the protein content of the rainbow trout’s formulated feeds were HIL0 (455.9 g kg^−1^), HIL25 (449.3 g kg^−1^), HIL50 (444.2 g kg^−1^), ImHIL25 (449.5 g kg^−1^), and ImHIL50 (447.0 g kg^−1^), indicating isonitrogenous feeds. Fat content was 153.1, 157.9, 177.9, 158.3, and 177.0 g kg^−1^ for HIL0, HIL25, HIL50, ImHIL25, and ImHIL50, respectively. Fat content was higher in feed with a higher HIL replacement rate for fishmeal (Table 2). Carbohydrate contents were HIL0 (184.4 g kg^−1^), HIL25 (189.6 g kg^−1^), HIL50 (145.0 g kg^−1^), ImHIL25 (155.6 g kg^−1^), and ImHIL 50 (147.8 g kg^−^^1^), respectively. The feeds produced in this study were formulated to have a protein:fat:carbohydrate ratio of 45:15:18. The protein:fat:carbohydrate ratio in the five diets produced in this study was HIL0 (46:15:18), HIl25 (45:16:18), HIL50 (44:18:15), ImHIL25 (45:16:16), and ImHIL50 (45:18:15), and the average protein:fat:carbohydrate ratio was 45:17:17 (Table 2). Therefore, similar to previous studies [19,31], the protein:fat:carbohydrate ratio of the formulated feed in the present study somewhat differed from that envisaged by formulation, albeit being almost similar.

### 3.3. Effect of ImHIL Intake on the Somatic Index of O. mykiss

The effect of the diets formulated in the current study on rainbow trout growth and immunity was evaluated for 6 weeks after feeding. The feeding rate (FR) was set at 2% of the fish’s weight per day when the fish weighed less than 200 g and 1.8% when the fish weighed over 200 g [37,38]. The rainbow trout were cultured at 18 °C. *O. mykiss* develops in the egg, fry, juvenile, and adult stages, and juveniles can be divided into parr and smolt [38]. The *O. mykiss* used in the present study was at the parr stage. The initial body weight (IBW) was 174.45 ± 8.95 g, and the initial length (IL) was 24.63 ± 0.43 cm (Table 3). To evaluate the effect of ImHIL on the survival rate, feed efficiency, and growth of *O. mykiss*, we measured survival rate, WG, SGR, FCR, HSI, and obesity after feeding for 3 and 6 weeks with feed containing ImHIL (Table 3). The survival rate was 100% in all groups, except the HIL25 group (96.7%).

At week 3, WG was 16.12, 15.75, 14.51, 13.87, and 12.82% in HIL0, HIL25, HIL50, ImHIL25, and ImHIL50, respectively. At week 6, WG was 32.85, 37.10, 33.73, 33.52, and 32.51% in HIL0, HIL25, HIL50, ImHIL25, and ImHIL50, respectively (Table 3). The WG in groups fed HIL- and ImHIL-containing feed was significantly lower than in that of the HIL0 group after 3 weeks. However, after 6 weeks, the HIL25, HIL50, ImHIL25, and ImHIL50 groups showed similar WG compared with the HIL0 group. In the previous study, the WGs in Eurasian sea bass [21] and rainbow trout [39] fed with the feed on which part of fishmeal is replaced with HIL were similar to those of only fishmeal feed, and this is consistent with the result in the HIL group in this study. Also, in this study, at week 6 after feeding, the WG of the ImHIL group was similar to the results of the HIL and fishmeal feed groups.

The specific growth rate (SGR) represents the daily growth of fish during the experiment. The SGR of rainbow trout fed the five diets for 6 weeks were 0.676, 0.745, 0.692, 0.688, and 0.668% in HIL0, HIL25, HIL50, ImHIL25, and ImHIL50, respectively. SGR was significantly higher in the HIL25, HIL50, and ImHIL25 groups (Table 3). In addition, SGRs in all HIL- and ImHIL groups were higher than HIL0. Through the results of previous studies as well as survival rate, WG, and SGR in this study [20,38,39,40], we confirmed that ImHIL had no effect on survival or weight loss in rainbow trout, so it can be concluded that AMPs contained in ImHIL are not toxic to rainbow trout. Therefore, their powders could be used instead of fishmeal.

The feed coefficient is the amount of feed needed for fish to grow by 1 g, and the reciprocal of the feed coefficient multiplied by 100 gives the feed conversion ratio (FCR). At week 3, FCR was 76.68, 76.60, 71.14, 68.40, and 59.15% in HIL0, HIL25, HIL50, ImHIL25, and ImHIL50, respectively. At week 6, FCR was 44.49, 59.36, 52.99, 54.51, and 51.60% in HIL0, HIL25, HIL50, ImHIL25, and ImHIL50, respectively. The FCR of O. mykiss fed HIL-containing feed was highest in HIL0 and lowest in ImHIL50 at week 3. At week 6, the FCR was higher in the HIL and ImHIL groups than in HIL0 (Table 3). There was no significant difference between the HIL and ImHIL feed groups. Because of the decrease in FR, the overall FCR appears to be lower in week 6 than in week 3.

The hepatosomatic index (HSI) is the ratio of liver to body weight. The higher the HSI, the healthier the fish. Accordingly, HSI is used as a measure of fish health [41]. At week 3, the HSI was 0.83, 0.91, 0.83, 0.85, and 0.77 in HIL0, HIL25, HIL50, ImHIL25, and ImHIL50, respectively. At week 6, HSI was 0.86, 0.95, 0.86, 0.98, and 0.89 in HIL0, HIL25, HIL50, ImHIL25, and ImHIL50, respectively. Although HSI levels were significantly higher in the HIL feed group than in the ImHIL group, the HSI growth rate between week 3 and 6 was higher in the ImHIL25 (15.29%) and ImHIL50 (15.58%) fed groups than HIL0 (3.61%), HIL25 (4.40%), and HIL50 (3.61%). This is likely attributed to the various AMPs in ImHIL, which increase the nutrient absorption rate by increasing intestinal villi length in the vertebrate intestine [27]. Therefore, it is speculated that even if the same amount of nutrients was ingested, the nutrient absorption rate in the ImHIL feed group with increased AMPs was higher. Since the liver is an energy reservoir that plays an important role in energy metabolism [42], it can be assumed that the higher the rate of nutrient absorption, the higher HSI growth rate. From the results of HSI growth rate, we found that ImHIL, which contains a large amount of AMP, has a growth-promoting effect by promoting nutrient absorption in rainbow trout.

Next, the effect of ImHIL on obesity in *O. mykiss* was assessed. Obesity is the ratio of fish weight to length, with a normal range of 1.10 to 1.20; a ratio less than 1.10 indicates malnutrition [43]. At week 3, obesity was 1.14, 1.18, 1.18, 1.19, and 1.19 in HIL0, HIL25, HIL50, ImHIL25, and ImHIL50, respectively. At week 3, the obesity of *O. mykiss* fed HIL- or ImHIL-blended feeds was significantly higher than that of HIL0. At week 6, obesity in the HIL0 and ImHIL50 groups was the highest as 1.18. In all groups, obesity was 1.14–1.19, indicating normal obesity (Table 3).

### 3.4. Effect of ImHIL Intake on the Hematological Index of O. mykiss

The hematological parameters are useful tools for monitoring fish growth and nutritional metabolism in response to dietary supplementation [44,45]. To evaluate the effect of ImHIL on *O. mykiss* growth, we measured RBC count, PCV, and IGF-1 content in rainbow trout after feeding diets containing ImHIL for 3 and 6 weeks (Figure 2).

The normal RBC count in rainbow trout is 0.71–1.73 × 10^6^ mm^−3^ [46]. At week 3, RBC count was 1.01, 0.87, 0.98, 0.99, and 1.06 × 10^6^ mm^−3^ in HIL0, HIL25, HIL50, ImHIL25, and ImHIL50, respectively. At week 6, RBC count was 0.85, 0.72, 0.85, 0.80, and 0.97 × 10^6^ mm^−3^ in HIL0, HIL25, HIL50, ImHIL25, and ImHIL50, respectively. The results of Duncan’s multiple range tests (*p* < 0.05) showed no significant differences among the groups, including HIL0, by the third week after feeding the formulated diets. At week 6, the RBC count was highest in the ImHIL50 group, and the HIL50 group had a higher RBC count than that of the HIL25 group. The RBC counts of all groups were within the normal range, the RBC count at week 6 did not increase compared with week 3 (Figure 2A). When peppermint, coriander, and curcumin, which have antibacterial activity, were fed to Caspian whitefish and rainbow trout, the RBCs increased in a concentration-dependent manner [44,45,47]. This increased number of RBCs promotes cellular respiration due to increased hemoglobin and promotes metabolism, thereby bringing about a growth-promoting effect [48]. When oral administration of feed additives with antibacterial activity, enteric pathogens are reduced due to their antibacterial activity, so nutrients absorbed by pathogens can be further absorbed into the fish body [44,45]. In general, HIL contains a small number of AMPs because it is in constant contact with microorganisms during ingestion and respiration, but ImHIL contains much more AMPs due to bacterial injection. For this reason, it was speculated that the ImHIL50 group showed the highest increase in the RBC count. On the other hand, since chitinase is present in the intestine of rainbow trout, chitin, which constitutes the exoskeleton of HIL, is decomposed into glucosamines, and the glycoprotein such as erythropoietin biosynthesized by these affects the increase in the number of RBCs [49]. For this reason, it is inferred that the RBC count increased in a concentration-dependent manner in the HIL and ImHIL groups.

The normal PCV in rainbow trout is 24–55% [46]. The PCVs in all feed groups were normal (Figure 2A). In the case of hematocrit, PCV was 38.93, 30.76, 35.05, 36.30, and 32.51% in HIL0, HIL25, HIL50, ImHIL25, and ImHIL50 at week 3, respectively. The PCV of HIL0 was higher than in that of other groups in week 3 (*p* < 0.05). At week 6, PCV was 38.00, 30.29, 38.97, 35.98, and 38.20% in HIL0, HIL25, HIL50, ImHIL25, and ImHIL50, respectively. Therefore, the other groups, except HIL25, were similar to the HIL0.

IGF-1 plays a vital role in promoting cell proliferation and differentiation during the growth and development of juvenile *O. mykiss* [50]. At both week 3 and 6 after feeding, we observed the highest IGF-1 content in the ImHIL50-fed group (Figure 2B). High IGF-1 levels in ImHIL were thought to be associated with increased nutrient absorption and storage due to the increased AMPs (Table 3). Therefore, feed containing ImHIL is expected to have a beneficial effect in promoting the growth and development of rainbow trout.

Among somatic and hematological indicators, WG, SGR, and FCR were superior to HIL- and ImHIL diets compared to HIL0, and HIL and ImHIL were similar. Further, the RBC count and IGF-1 showed the highest value in the ImHIL50 group. Therefore, as in the previous results [19,20,21], it was confirmed that the HIL-added feed showed a better growth-promoting effect than only fishmeal-based feed, and the ImHIL feed containing a large amount of AMPs showed a better growth-promoting effect without toxicity (Figure 2).

### 3.5. Effects of ImHIL Intake on Immunity Enhancement of O. mykiss

#### 3.5.1. Analysis of AST in the Blood of *O. mykiss* Fed with ImHIL

The fish liver is an important organ directly affected by food, pollutants, toxins, parasite, and microorganisms, and is also an important immunological marker because it is also involved in host defense [51]. As immunological functions, the liver executes nonspecific phagocytosis, produces acute-phase proteins and complements, and removes pathogens [52]. AST is an enzyme in hepatocytes used as an indicator of liver-related diseases since its concentration increases when cells are damaged [53]. Accordingly, we determined the AST in *O. mykiss* blood after the feeding period to evaluate the effect of ImHIL-blended feeds on *O. mykiss*’ health. The initial AST content of the rainbow trout was 67.57. At week 3, AST content in HIL0, HIL25, HIL50, ImHIL25, and ImHIL50 was 31.90, 37.44, 38.67, 33.78, and 29.7, respectively. At week 6, AST content was 30.57, 28.58, 33.16, 23.78, and 27.97 in HIL0, HIL25, HIL50, ImHIL25, and ImHIL50, respectively. We observed decreased AST levels in all groups at week 3 and 6 compared to that of the initial level. At week 3, although AST values were statistically similar among groups, they were highest in HIL50 and lowest in ImHIL50. At week 6, ImHIL25 had the lowest AST level in the group, and ImHIL50 was comparable to HIL0 and HIL25 (Figure 3A). As corroborated by these findings, we expected ImHIL to enhance health by minimizing damage to rainbow trout liver cells. AST as well as WG, FCR, SGR, obesity, RBC, and PCV can be useful indicators for indirectly assessing the physiological well-being and health of fish [54,55]. A high AST value indicates physiological problems of a nutritional or environmental nature as AST is secreted following hepatocyte destruction. In this study, the AST value of the HIL-containing diet group was similar to those of HIL0, which is consistent with the previously reported results [34]. At week 6, ImHIL had lower AST values, respectively, compared to the HIL group that replaced the same content of a fish meal, which is thought to be due to AMPs, which is the main difference between the ImHIL and HIL groups. ImHIL may have a hepatocyte-protective effect [22].

#### 3.5.2. Analysis of lysozyme Activity in the Blood of *O. mykiss* Fed with ImHIL

Lysozyme is an enzyme that destroys the cell wall of bacteria and plays an important role in the host’s immune defense [56]. At week 3, lysozyme concentration was 568.78, 648.32, 659.90, 691.55, and 732.58 U L^−1^ in the blood of HIL0, HIL25, HIL50, ImHIL25, and ImHIL50, respectively. At week 6, lysozyme concentration was 610.10, 683.83, 804.27, 875.29, 923.16 U L^−1^ in HIL0, HIL25, HIL50, ImHIL25, and ImHIL50, respectively. Comparison of lysozyme levels at weeks 3 and 6 indicated that lysozyme levels significantly increased in the ImHIL25 and ImHIL50 groups (Figure 3B). In previous studies, an increase in lysozyme activity was reported to enhance the protective efficacy against foreign pathogen infection [57]. Since lysozyme is produced and secreted by the liver [57], it is speculated that lysozyme levels were higher in the ImHIL-containing diet group, which had lower AST levels than in the other groups. It was previously reported that the lysozyme activity of skin mucus was significantly increased in Nile tilapia fed HIL-containing diet [58]. In addition, lysozyme and phagocytic activity were increased when fed to European perch by replacing 50% of the fishmeal with HIL [21]. Therefore, like the results previously reported, the lysozyme activity of the HIL group was significantly higher than that of HIL0 in this study. Lysozyme level in the ImHIL group was significantly higher than that of HIL, and it is inferred that this is because, unlike HIL, ImHIL contains various overexpressed AMPs. Therefore, ImHIL may contribute to increasing the defense against foreign substances of O. mykiss.

#### 3.5.3. Analysis of IgM in the Blood of *O. mykiss* Fed with ImHIL

IgM is the first antibody produced upon infection with an external pathogen. Since it is expressed first by external pathogens, it is an important immune indicator [59,60]. At week 3, IgM was 2.74, 1.94, 2.17, 1.88, 1.87 ng mL^-1^ in HIL0, HIL25, HIL50, ImHIL25, and ImHIL50, respectively; at week 6 it was 2.84, 1.83, 2.25, 2.04, 1.93 ng mL^-1^ in HIL0, HIL25, HIL50, ImHIL25, and ImHIL50, respectively (Figure 3C). In a previous study, when feed containing guava leaf powder was fed to Labeo rohita, there was an increase in lysozyme but no change in IgM [61]. Even when feed contained probiotics, IgM increased until 30 days; nevertheless, 60 days later, there was a decrease as much as the initial value, suggesting that the increase in IgM production is a short-term phenomenon attributable to the presence of immunostimulants [61,62]. IgM is an antibody expressed when there is external pathogen invasion and is an adaptive immune system mechanism that causes B cell maturation or T lymphopoiesis [59,63]. IgM is a substance at the forefront of recognizing that foreign substances invaded the body, and their increased expression is a signal that foreign substances entered the body and were detected. In the HIL and ImHIL-fed groups, IgM content was lower than in that of the HIL0 group. These results are considered to be because rainbow trout living in freshwater have eaten insects ecologically [64], and thus not only did not recognize HIL, an insect, as a foreign substance, but also did not consider the AMPs contained in ImHIL to be harmful. In addition, AMPs are safe in humans [65]; therefore, we expect that ImHIL could safely replace fishmeal as a feed ingredient in fish farming where disease susceptibility is higher than in that of natural ecosystem environments [66].

## 4. Conclusions

ImHIL extracts, in which the expression of various AMPs was induced, showed antibacterial activity against strains causing fish diseases. In the ImHIL group, all growth indicators used in the current study fell into the normal category. Based on the results of survival rate, body weight, HSI and IgM analysis, ImHIL did not act as a toxic substance to rainbow trout. Therefore, we verified the possibility of using ImHIL as a novel feed ingredient. In particular, in the ImHIL group, growth promotion and immunity-enhancing effects were superior to those of HIL0 and other HIL groups, as evidenced by significant increases in RBC count and IGF-1 as growth indicators, as well as a decrease in AST and increased lysozyme as immune indicators. Therefore, we propose that ImHIL could be of high value as a novel feed ingredient that can improve the health and economics of rainbow trout aquaculture production.

## Figures and Tables

**Figure 1 insects-12-00801-f001:**
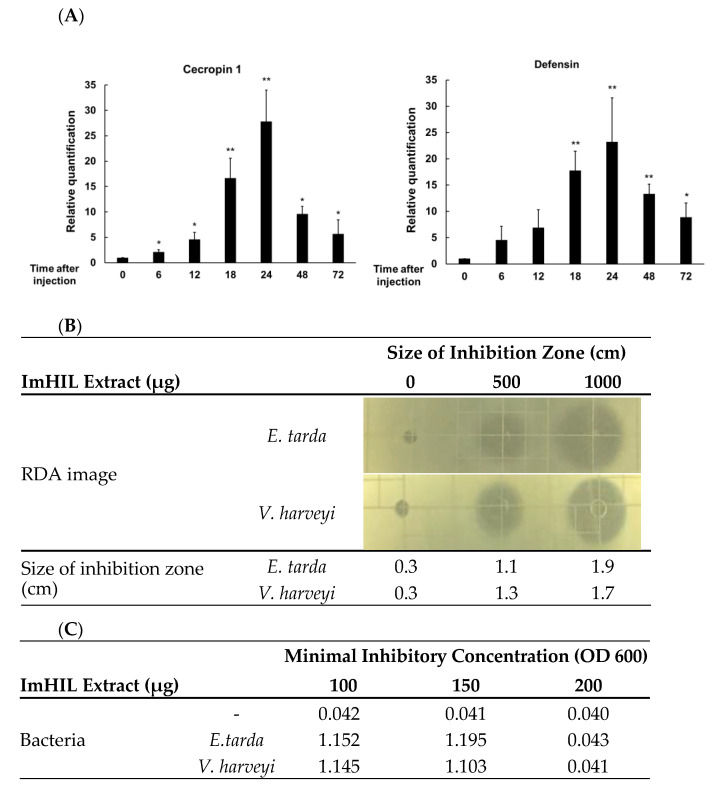
Analysis of cecropin 1 and defensin transcript (**A**), radial diffusion assay (**B**), and minimum inhibitory concentration (**C**) using ImHIL inoculated with *Lactobacillus plantarum*. ImHIL, immunized HIL. * *p* <0.05 and ** *p* <0.01 represent significant differences between 0 h and 2, 4, 8, 16, 24, 48, and 72 h postinjection, respectively.

**Figure 2 insects-12-00801-f002:**
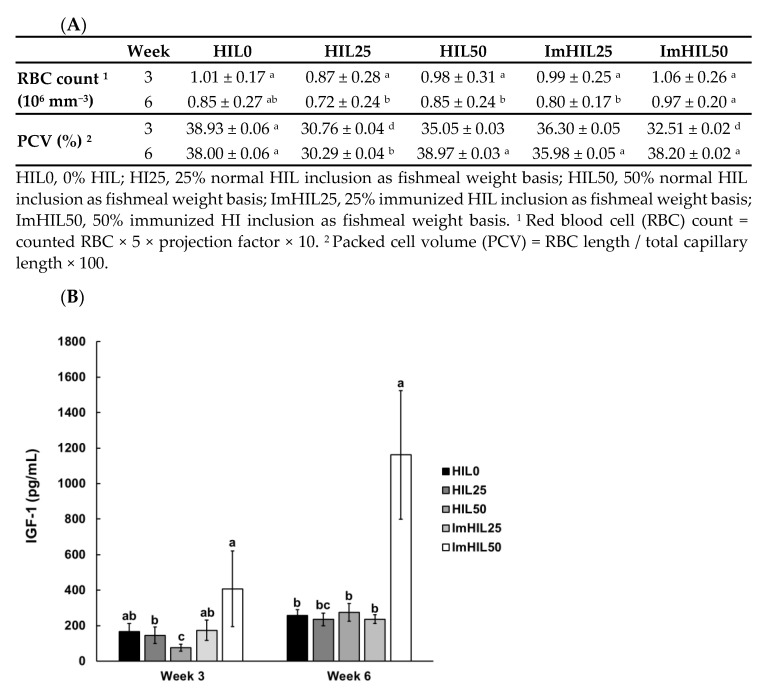
Hematological response of *O. mykiss* fed with five experimental diets. Red blood cell (RBC) count, packed cell volume (PCV) (**A**), and insulin-like growth factor-1 (IGF-1) level (**B**) in *O. mykiss* blood serum were measured. HIL0, 0% HIL; HI25, 25% normal HIL inclusion as fishmeal weight basis; HIL50, 50% normal HIL inclusion as fishmeal weight basis; ImHIL25, 25% immunized HIL inclusion as fishmeal weight basis; ImHIL50, 50% immunized HIL inclusion as fishmeal weight basis. Values were taken as a mean of triplicate analyses. Vertical error bars represent standard error of the mean, and lowercase alphabetic characters represent significant differences (*p* < 0.05) among groups at each week 3 and week 6 as determined by Duncan’s multiple range test, respectively (*n* = 9, 3 tanks, 3 fish per tank).

**Figure 3 insects-12-00801-f003:**
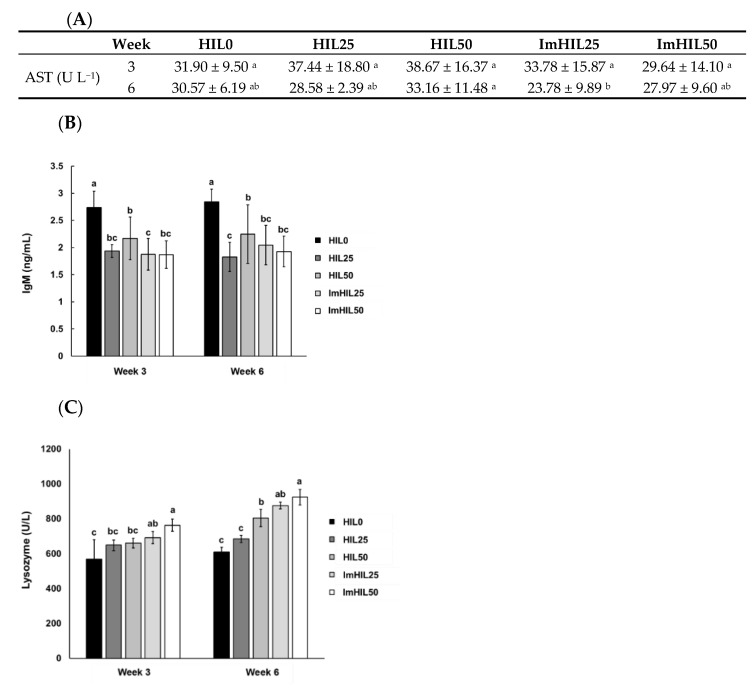
Measuring immunity-enhancing degree in rainbow trout fed with imHIL-containing feed. Aspartate aminotransferase (AST) (**A**), lysozyme activity (**B**), and immunoglobulin M (IgM) (**C**) were quantitatively analyzed. Initial AST level of rainbow trout blood serum was 67.57 ± 16.72 U L^−1^. Values are means ± standard deviations of triplicates. HIL0, 0% HIL; HI25, 25% normal HIL inclusion as fishmeal weight basis; HIL50, 50% normal HIL inclusion as fishmeal weight basis; ImHIL25, 25% immunized HIL inclusion as fishmeal weight basis; ImHIL50, 50% immunized HIL inclusion as fishmeal weight basis. Vertical error bars denote standard error of mean, and alphabetic characters represent significant differences (*p* < 0.05) among each group at week 3 and week 6 as determined by Duncan’s multiple range test, respectively (*n* = 9, 3 tanks, 3 fish per tank).

**Table 1 insects-12-00801-t001:** Primer sequences used to analyze HiCec1 (*Hermetia illucens* cecropin1) and HiDef1 (*H. illucens* defensin1) gene expressions using real-time PCR.

Name		Sequence
HiCec1 (Accession: JX997953)	Forward	5’-TTGGTCAACGAGTTCGTGATGC-3’
	Reverse	5’-TCCTTGTTGTGGTGGTCCACCT-3’
HiDef1 (Accession: KF805347)	Forward	5’-AGGTGGTGGAGCAGCATTAC-3’
	Reverse	5’-ACGACGTCCCAAAGCAATAC-3’
Act5C (Accession: P10987)	Forward	5′-AAGGACTCGTACGTGGGTG-3′
	Reverse	5′-CATCTTCTCACGGTTGGC-3′

**Table 2 insects-12-00801-t002:** Proximate composition of formulated rainbow trout feed. HIL0, 0% HIL; HIL25, 25% nonimmunized HIL replacement based on the weight of fishmeal; HIL50, 50% nonimmunized HIL replacement based on the weight of fishmeal; ImHIL25, 25% immunized HIL replacement based on the weight of fishmeal; ImHIL50, 50% immunized replacement based on the weight of fishmeal. DM, dry matter; CP, crude protein; EE, ether extract. The values were taken as mean of duplicate analysis.

	HIL0	HIL25	HIL50	ImHIL25	ImHIL50
Ingredients, g kg^−1^					
Fish meal	650	470	285	480	310
Fish oil	100	80	63	100	105
Noninduced HI	-	250	500	-	-
Immune induced HI	-	-	-	250	500
Wheat meal	200	150	102	120	35
Gelatinized starch (D500)	50	50	50	50	50
Mineral mixture	10	10	10	10	10
Vitamin Mixture	10	10	10	10	10
Proximate composition					
DM, g kg^−1^	943.5	940	937.4	936.9	938.3
CP, g kg^−1^	455.9	449.3	444.2	449.5	447.0
EE, g kg^−1^	153.1	157.9	177.9	158.3	177.0
Carbohydrate, g kg^−1^	184.4	189.6	145.0	155.6	147.8
Ash, g kg^−1^	132.4	130.7	140.7	126.6	147.8
Gross energy, cal g^−1^	4900.0	4899.0	4919.5	4808.0	4981.5

**Table 3 insects-12-00801-t003:** Growth performance, survival rate, and somatic indices of rainbow trout fed with experimental diets.

	Week	HIL0	HIL25	HIL50	ImHIL25	ImHIL50
IBW (g)	0	170.90 ±12.83 ^a^	178.44 ± 12.21 ^a^	176.45 ± 0.10 ^a^	177.54 ± 0.32 ^a^	167.71 ± 12.80 ^a^
IL (cm)	0	24.55 ± 0.68 ^a^	24.94 ± 0.14 ^a^	24.67 ± 0.21 ^a^	24.76 ±0.05 ^a^	24.05 ± 0.61 ^a^
SR (%)	3	100 ^a^	100 ^a^	100 ^a^	100 ^a^	100 ^a^
	6	100 ^a^	96.7 ^a^	100 ^a^	100 ^a^	100 ^a^
BW (g)	3	198.45 ± 0.27 ^ab^	206.54 ± 7.35 ^a^	202.05 ± 8.12 ^ab^	202.16 ± 4.63 ^ab^	189.21 ± 3.70 ^b^
	6	226.98 ± 2.25 ^a^	244.01 ± 10.19 ^a^	235.97 ± 24.12 ^a^	237.05 ± 33.68 ^a^	222.03 ± 17.11 ^a^
WG ^1^ (%)	3	16.12 ± 0.94 ^a^	15.75 ± 6.20 ^b^	14.51 ± 0.99 ^b^	13.87 ± 1.54 ^b^	12.82 ± 2.52 ^c^
	6	32.85 ± 0.86 ^a^	37.10 ± 7.93 ^a^	33.73 ± 5.68 ^a^	33.52 ± 6.10 ^a^	32.51 ± 3.69 ^a^
SGR ^2^ (%/day)	6	0.676 ± 0.002 ^c^	0.745 ± 0.001 ^a^	0.692 ± 0.003 ^b^	0.688 ± 0.001 ^b^	0.668 ± 0.001 ^c^
FCR ^3^ (%)	3	76.68 ± 10.21 ^a^	76.60 ± 24.18 ^a^	71.14 ± 5.05 ^a^	68.40 ± 7.72 ^b^	59.15 ± 6.75 ^b^
	6	44.49 ± 1.44 ^b^	59.36 ± 13.08 ^a^	52.99 ± 13.08 ^a^	54.51 ± 17.07 ^a^	51.60 ± 7.90 ^a^
HIS ^4^	3	0.83 ± 0.06 ^b^	0.91 ± 0.13 ^a^	0.83 ± 0.01 ^b^	0.85 ± 0.22 ^b^	0.77 ± 0.11 ^c^
	6	0.86 ± 0.12 ^b^	0.95 ± 0.22 ^a^	0.86 ± 0.06 ^b^	0.98 ± 0.30 ^a^	0.89 ± 0.09 ^b^
HGR ^5^ (%)		3.61 ± 0.08 ^b^	4.40 ± 0.14 ^b^	3.61 ± 0.07 ^b^	15.29 ± 0.21 ^a^	15.58 ± 0.08 ^a^
Obesity	3	1.14 ± 0.01 ^b^	1.18 ± 0.03 ^a^	1.18 ± 0.01 ^a^	1.19 ± 0.03 ^a^	1.19 ± 0.02 ^a^
	6	1.18 ± 0.02 ^a^	1.15 ± 0.08 ^b^	1.17 ± 0.01 ^a^	1.15 ± 0.01 ^b^	1.18 ± 0.01 ^a^

HIL0, 0% HIL; HI25, 25% normal HIL inclusion as fishmeal weight basis; HIL50, 50% normal HIL inclusion as fishmeal weight basis; ImHIL25, 25% immunized HIL inclusion as fishmeal weight basis; ImHIL50, 50% immunized HI inclusion as fishmeal weight basis; IBW, initial body weight; IL; SR, survival rate; initial length; BW, body weight; WG, weight gain; SGR, specific growth rate; FCR, feed conversion ratio; HSI, hepatosomatic index; HGR, HSI growth rate. Superscripted lowercase alphabetic characters represent significant differences (*p* < 0.05) among groups at each week 3 and week 6 as determined by Duncan’s multiple range test, respectively (*n* = 9, 3 tanks, 3 fish per tank). ^1^ WG (%) = [{Last weight (g) − initial weight (g)}/initial weight (g)] ×100. ^2^ SGR (%/day) = {(ln final weight − ln initial weight)/number of days} × 100. ^3^ FCR (%) = (weight gain/feed gain/feed intake) × 100. ^4^ HSI = (liver weight/body weight) × 100. ^5^ HSI growth rate = (week 6 HSI/week 3 HSI) × 100.

## Data Availability

Not applicable.

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
