# Peer review of "Effect of Feed Containing Hermetia illucens Larvae Immunized by Lactobacillus plantarum Injection on the Growth and Immunity of Rainbow Trout (Oncorhynchus mykiss)"

_insects, 2021, doi:10.3390/insects12090801_

Round 1

Reviewer 1 Report

It is interesting to study the effects of feeding BSF larvae immunized by the bacteria on the immunity physiology and growth of rainbow trout.

Major problems

There is no description of the sex and maturation of the rainbow trout that were used. Their sex and maturity have important effects on the immunity、physiology, and growth of fishes.  A detailed description of the sex of these fish would be very important. Also, in data analysis, males and females must be analyzed and evaluated separately. If there is a gender bias, the effect of gender cannot be ignored.

The authors discuss the significant increase in IGF-1 in the ImHIL50 group in relation to growth, but the actual experimental results show no effect on fish growth. It is known that Igf-1 has many different functions. Isn't it dangerous to simply consider its relationship to growth?

In the case of fish, HIS is greatly affected by sex and maturity. Since the sex and maturity used are not listed, the discussion of HIS is meaningless.

Others

The number of tanks and the number of fishes are described, but the number of tanks for each experimental group is not recognized.  Are all experimental groups duplicitous or triplicate?

The IGF-1 assay kit is not described in detail; Cusabio's IGF-1 assay kit is sold "for fish", but please describe the cross-activity for rainbow trout. IgM assay kit has also same problem.

Although positive observations have been made about AST data, these observations about AST are meaningless because no statistically significant data have been presented.

Problems of Figures and Table

In Fig. 1, there are no symbols to indicate each picture, which is not understandable. Isn't this figure necessary? The explanation of Fig. 2 (B) is insufficient. Also, there is a problem with the placement of the pictures.

There is no information about standard deviation or error in the values in Table 3. It would be easier to understand if they were mentioned.

Author Response

Thank you for your time in handling our manuscript. We have improved our manuscript according to the reviewers' suggestions. The response to the specific comments is as follows:

<Revision List>

# Reviewer: 1

Comments and Suggestions for Authors

It is interesting to study the effects of feeding BSF larvae immunized by the bacteria on the immunity physiology and growth of rainbow trout.

Major problems

  1. There is no description of the sex and maturation of the rainbow trout that were used. Their sex and maturity have important effects on the immunity, physiology, and growth of fishes. A detailed description of the sex of these fish would be very important. Also, in data analysis, males and females must be analyzed and evaluated separately. If there is a gender bias, the effect of gender cannot be ignored.

Answer: The rainbow trout used in our experiment was in the parr stage. Parr belongs to the juvenile period, and it is difficult to distinguish between the sexes because the sex glands do not develop at this time [1]. In addition, the rainbow trout used is a triploid with suppressed gonad development and is an individual whose gonads do not mature even when they become adults. In general, triploid rainbow trout has a higher growth rate than diploid, so it is mainly used in aquaculture [2]. Therefore, since this study was conducted with the purpose of applying the immunized Hermetia illucens to feed for aquaculture, triploid rainbow trout, which is mainly used for aquaculture, was used. Following the reviewer's suggestion, the developmental stages of the triploid rainbow trout used in this study were added to Materials and Methods (line 153).

  1. The authors discuss the significant increase in IGF-1 in the ImHIL50 group in relation to growth, but the actual experimental results show no effect on fish growth. It is known that Igf-1 has many different functions. Isn't it dangerous to simply consider its relationship to growth?

Answer: In lines 383-388, it is described as “Among somatic and hematological indicators, WG, SGR, and FCR were superior to HIL- and ImHIL diets compared to HIL0, and HIL and ImHIL were similar. And, the RBC count and IGF-1 showed the highest value in the ImHIL50 group. Therefore, as in the previous results, it was confirmed that the HIL-added feed showed a better growth-promoting effect than only fishmeal-based feed, and the ImHIL feed containing a large amount of AMPs showed a better growth-promoting effect without toxicity”. Although IGF-1 has various functions, it has been used as a growth indicator among them. In previous studies, it was used as a growth index to confirm the growth performance of rainbow trout, Siberian sturgeon, and snapper [3-7].

  1. In the case of fish, HSI is greatly affected by sex and maturity. Since the sex and maturity used are not listed, the discussion of HSI is meaningless.

Answer: The rainbow trout used in our experiment was in the parr stage. Parr belongs to the juvenile period, and it is difficult to distinguish between the sexes because the sex glands do not develop at this time [1]. In addition, the rainbow trout used is a triploid with suppressed gonad development and is an individual whose gonads do not mature even when they become adults. In general, triploid rainbow trout has a higher growth rate than diploid, so it is mainly used in aquaculture [2]. Therefore, since this study was conducted with the purpose of applying the immune-induced Hermetia illucens to feed for aquaculture, triploid rainbow trout, which is mainly used for aquaculture, was used. Following the reviewer's suggestion, the developmental stages of the triploid rainbow trout used in this study were added to “Materials and Methods” (line 153).

Others

  1. The number of tanks and the number of fishes are described. But the number of tanks for each experimental group is not recognized. Are all experimental groups duplicitous or triplicate?

Answer: We treated 5 types of artificial diets in triplicate in this study. Therefore, the total number of tanks was 15. This content was written in the legends of the Figure and Table (line 327, 405, 489), but has been omitted from the text. However, as suggested by the reviewer, we added the number of iterations to “Materials and Methods” (lines 154-156).

  1. The IGF-1 assay kit is not described in detail: Cusabio’s IGF-1 assay kit is sold “for fish”, but please describe the cross-activity for rainbow trout. IgM assay kit has also same problem.

Answer: Cusabio’s IGF-1 assay kit was used to quantify IGF-1 in various species such as smolt [3] and juvenile [4] stage rainbow trout, as well as carp fish [8] and marbled flounder [9]. IgM assay kit used in this study has also been used to measure immune markers in fish such as rainbow trout [10], eel [11], gibel carp [12], and Nile tilapia [13]. Therefore, we selected the kit by referring to the above references. We also added details about the assay kit based on the reviewer’s comment in lines 188-190, 201-203.

  1. Although positive observations have been made about AST data, these observations about AST are meaningless because no statistically significant data have been presented.

Answer: At week 6, ImHIL had lower AST values, respectively, compared to the HIL group that re-placed the same content of a fish meal. This result was significant by Duncan's multiple range test. Therefore, we changed the contents of lines 430-431 from “The ImHIL group exhibited a lower AST value than the HIL groups,” to “At week 6, ImHIL had lower AST values, respectively, compared to the HIL group that replaced the same content of a fish meal,”.

Problem of Figures and Table

  1. In Fig. 1, there are no symbols to indicate each picture, which is not understandable. Isn’t this figure necessary? The explanation of Fig. 2 (B) is insufficient. Also, there is a problem with the placement of the pictures.

Answer: In the case of Fig. 1, it was intended to indicate that all the manufactured feeds were similar in shape, but there is no problem in understanding it even if I delete it. Following the reviewer’s suggestion, Fig. 1 was deleted. In addition, the position of the pictures in Fig. 2 was corrected (line 230).

  1. There is no information about standard deviation or error in the values in Table 3. It would be easier to understand if they were mentioned.

Answer: According to the reviewer’s suggestion, each standard deviation was added to Table 3.

<References>

  1. William, S.H. Smolt transformation: evolution, behavior, and physiology. Fish. Res. Board Can. 1976, 33 (5), 1233–1252.
  2. Kim, D.S.; Kim, I.B.; Baik, Y.G. Growth and gonadal development of triploid rainbow trout, Salmo gairdneri. J. Aquacul. 1988. 1, 45-51.
  3. Hou, Z.S.; Wen, H.S.; Li, J.F.; He, F.; Li, Y.; Qi, X. Environmental Hypoxia Causes Growth Retardation, Osteoclast Differentiation and Calcium Dyshomeostasis in Juvenile Rainbow Trout (Oncorhynchus mykiss). Total Environ. 2020, 25, 705: 135272.
  4. Bernard, B. A temperature shift on the migratory route similarly impairs hypo-osmoregulatory capacities in two strains of Atlantic salmon (Salmo salar) smolts. Fish Physiol. Biochem. 2019, 45(4), 1245-1260.
  5. Babaei, S.; Kenari, A.A.; Hedayati, M.; Sadati, M.A.Y. Growth response, body composition, plasma metabolites, digestive and antioxidant enzymes activities of Siberian sturgeon (Acipenser baerii, Brandt, 1869) fed different dietary protein and carbohydrate: lipid ratio. Aquac. Res. 48(6), 2642-2654.
  6. Amri, A.; Kessabi, K.; Bouraoui, Z.; Sakli, S.; Gharred, T.; Guerbej, H.; Messaoudi, I.; Jebali, J. Effect of melatonin and folic acid supplementation on the growth performance, antioxidant status, and liver histology of the farmed gilthead sea bream (Sparus aurata L.) under standard rearing conditions. Fish Physiol. Bioc 2020, 46(6), 2265-2280.
  7. Moriyama, S.; Ayson, F.G.; Kawauchi, H. Growth regulation by insulin like growth factor-1 in fish. Biotechnol. Biochem. 2000, 64 (8), 1553–1562.
  8. Korkmaz, N.; Orun, I. The effects of pesticide neemazal on the growth hormone (GH) and insulin-like growth hormone (IGF-1) of carp fish (CyprinusTihomir). Technol. 2016, 3(1).
  9. Cho, J.H.; Lee, S.; Lee, B.J.; Hur, S.W.; Kim, K.W.; Son, M.H.; Yoo, D.Y. A preliminary study of dietary protein requirement of juvenile marbled flounder (Pseudopleuronectes yokohamae). Nutr. 2021, 7(2), 548-555.
  10. Uluky G.; Metin, S.; Kubilay, A.; Guney, S.; Yildirim, P.; Seydim, Z.; Tas, T.K.; Gumus, E. The effect of kefir as a dietary supplement on nonspecific immune response and disease resistance in juvenile rainbow trout, Oncorhynchus mykiss (Walbaum 1792). World Aquac. Soc. 2016, 48(2), 248-256.
  11. Ulukoy, G.; Baba, E.; Ontas, C. Effect of oyster mushroom, Pleurotus ostreatus, extract on hemato-immunological parameters of rainbow trout, Oncorhynchus mykiss World Aquac. Soc. 2016, 47(5), 676-684.
  12. Zhang, P.; Cao, S.; Zou, T.; Han, D.; Liu, H.; Jin, J.; Yang, Y.; Zhu, X.; Xie, S.; Zhou, W. Effects of dietary yeast culture on growth performance, immune response and disease resistance of gibel carp (Carassius auratus gibelio CAS Ⅲ). Fish Shellfish Immunol. 2018, 82, 400-407.
  13. El-Gawad, E.A.; Asely, A.M.E.; Soror, E.I.; Abbass, A.A.; Austin, B. Effect of dietary Moringa oleifera leaf on the immune response and control of Aeromonas hydrophila infection in Nile tilapia (Oreochromis niloticus). Intern. 2020, 28, 389-402.

Reviewer 2 Report

Very interesting and innovating research that contibutes to the sustainability of the aquaculture sector. In addition, also the insect rearing sector can generate more added value to their produce. The paper is easy readable and spelling and sentence construction are adequate.

In what follows some minor suggestions:

L 140-145: What about carbohydrate and ash analyses?

L 154: It could be interesting to provide a bit more information about the natural food of the species. For example, are they insectivorous?

L 178: "a 200-times dilutions": remove the "s" at the end of dilutions

L 322: Can you explain why you didn't measure the villi lengths to be sure? Could be interesting, but on the other hand I see that you already investigated a lot of indicators. 

L 493: "Measuring the degree of the immunity-enhancing degree in rainbow trout": ommit "the degree of" because you already have "degree" after "immunity-enhancing"

L 515: "that ImHIL has of high value as a novel feed ingredient": replace "has" by "could be"

Author Response

Thank you for your time in handling our manuscript. We have improved our manuscript according to the reviewers' suggestions. The response to the specific comments is as follows:

<Revision List>

# Reviewer: 2

Comments and Suggestions for Authors

Very interesting and innovating research that contributes to the sustainability of the aquaculture sector. In addition, also the insect rearing sector can generate more added value to their produce. The paper is easy readable and spelling and sentence construction are adequate.

In what follows some minor suggestions:

  1. L 140-145: What about carbohydrate and ash analyses?

Answer: Following the reviewer's suggestion, carbohydrate and ash analyses were added to “Materials and Methods” (line 144-150).

  1. L 154: It could be interesting to provide a bit more information about the natural food of the species. For example, are they insectivorous?

Answer: In line 363-366, we mentioned “On the other hand, since chitinase is present in the intestine of rainbow trout, chitin, which constitutes the exoskeleton of HIL, is decomposed into glucosamines, and the glycoprotein such as erythropoietin biosynthesized by these affects the increase in the number of RBCs [50]”. This indirectly explains the insectivorous nature of rainbow trout. In addition, we also mentioned “These results are considered to be because rainbow trout living in freshwater have eaten insects ecologically [65], and thus not only did not recognize HIL, an insect, as a foreign substance but also did not consider the AMPs contained in ImHIL to be harmful.” in line 468-471.

  1. L 178: “a 200-times dilutions”: remove the “s” at the end of dilutions.

⟶ Answer: According to the reviewer’s suggestion, we corrected “a 200-times dilutions” to as “a 200-times dilution” (line 177).

  1. L 322: Can you explain why you didn’t measure the villi lengths to be sure? Could be interesting, but on the other hand I see that you already investigated a lot of indicators.

Answer: In this study, we confirmed the growth indicators such as weight gain, specific growth rate, feed conversion ratio, hepatosomatic index, obesity, red blood cell count, packed cell volume, and insulin like growth factor-1 of rainbow trout according to the feed treatment groups. Since the previous paper reported the association between AMPs and intestinal villi and growth (line 66-68), the relationship between the increase in growth indicators and villi in the ImHIL50 group containing a lot of AMPs was discussed (line 309-311).

  1. L 492: “Measuring the degree of the immunity-enhancing degree in rainbow trout”: omit “the degree of” because you already have “degree” after “immunity-enhancing”.

Answer: According to the reviewer’s suggestion, we corrected “Measuring the degree of the immunity-enhancing degree in rainbow trout” to as “Measuring the immunity-enhancing degree in rainbow trout” (line 479).

  1. L 515: “that ImHIL has of high value as a novel feed ingredient”: replace “has” by “could be”.

Answer: According to the reviewer’s suggestion, we corrected “that ImHIL has of high value as a novel feed ingredient” to as “that ImHIL could be of high value as a novel feed ingredient” (line 501).
